# Does Measles, Mumps, and Rubella (MMR) Vaccination Protect against COVID-19 Outcomes: A Nationwide Cohort Study

**DOI:** 10.3390/vaccines10111938

**Published:** 2022-11-16

**Authors:** Epiphane Kolla, Alain Weill, David Desplas, Laura Semenzato, Mahmoud Zureik, Lamiae Grimaldi

**Affiliations:** 1EPI-PHARE (Scientific Interest Group in Epidemiology of Health Products), French National Agency for the Safety of Medicines and Health Products, French National Health Insurance, CEDEX, 93285 Saint-Denis, France; 2INSERM (National Institute of Health and Medical Research), University of Paris-Saclay, University Versailles Saint Quentin, Anti-Infective Evasion and Pharmacoepidemiology Team, 78180 Montigny-Le-Bretonneux, France; 3Clinical Research Unit AP-HP, Paris-Saclay, Hôpital Raymond Poincare, School of Medicine Simone Veil, University Versailles Saint Quentin—University Paris Saclay, INSERM (National Institute of Health and Medical Research), CESP, Anti-Infective Evasion and Pharmacoepidemiology Team, 78180 Montigny-Le-Bretonneux, France

**Keywords:** MMR vaccine, SNDS, vaccination, COVID-19, hospitalizations

## Abstract

Cross-protection from previous live attenuated vaccines is proposed to explain the low impact of COVID-19 on children. This study aimed to evaluate the effect of live attenuated MMR vaccines on the risk of being hospitalized for COVID-19 in children. An exposed (MMR vaccine)–non-exposed cohort study was conducted using the nationwide French National Health Data System (SNDS). We included children born between 1 January 2009 and 31 December 2019. Exposure was defined as a claim of at least one dose of MMR vaccine since birth. Hospitalization for COVID-19 was defined using main diagnostic ICD10 codes. Non-conditional logistic regression was used to calculate the adjusted odds ratios (aORs) of the association between MMR exposure and hospitalization for COVID-19, controlling for socio-demographic and socio-economic factors, co-morbidities, and general health. In total, 6,800,542 (median age 6 IQR [3–8] years) children exposed to a MMR vaccine and 384,162 (6 [3–9] years) not exposed were followed up with for 18 months. Among them, 873 exposed to the MMR vaccine and 38 who were not exposed were hospitalized for COVID-19. In a multi-variate analysis, the exposure of children to MMR vaccination was not associated with a decreased risk of COVID-19 hospitalization versus non-exposure (aOR (95%CI) = 1.09 [0.81–1.48]). A stratified analysis by age showed an aOR = 1.03 [0.64–1.66] for children aged 1–4, an aOR = 1.38 [0.82–2.31] for those aged 5–9, and an aOR = 1.11 [0.54–2.29] for those aged 10–12. Our study suggests that the live attenuated MMR vaccine does not protect children against COVID-19 hospitalization.

## 1. Introduction

In December 2019, severe pneumonia cases were discovered in Hubei province in Wuhan (China). The cause of these cases was associated with a new virus, severe acute respiratory syndrome coronavirus-2 (SARS-CoV-2) [1]. The virus spread rapidly worldwide, with over 200 million people infected by 31 December 2021 (ECDC COVID-19 situation update worldwide). During the same period, France registered over ten million cases. The people most at risk and likely to be infected or to have severe forms of the disease are adults and the elderly with vulnerability factors [2]. Children have consistently been the least affected population, relative to adults, for severe forms [3,4,5].

One biological explanation discussed in the literature is the increasing expression of the ACE2 gene (cellular receptors to which the coronavirus binds through protein S) with age [6,7]. Another hypothesis is the cross-protection provided by previous live attenuated vaccines. The Bilié Calmette–Guérin (BCG) and measles–mumps–rubella (MMR) vaccines have been the subject of several studies and clinical trials (NCT04333732) evaluating the protective effect of these vaccines against SARS-CoV-2 infection [8,9]. These vaccines have been proposed based on their ability to stimulate innate immunity, which is non-specific for other diseases [10,11,12,13]. Low COVID-19 severity and a low incidence of mortality have been reported among MMR-vaccinated populations [14]. More recently, a study carried out among healthcare workers in Sweden reported a protective effect of MMR vaccines against SARS-CoV-2 infection in men only [15].

Providing scientific evidence that live attenuated vaccines have the potential to prevent infection and/or limit the severity of COVID-19 could have an impact on public health policies as the pandemic continues. It is, thus, important to know whether a live attenuated vaccine could be used to protect against SARS-CoV-2 infection and hospitalization for COVID-19, particularly in children. In France, MMR vaccine coverage has increased since 2018 [16]. We examined the hypothesis that this vaccine offers non-specific primary protection against COVID-19 by evaluating the effect of the live attenuated MMR vaccine on the risk of being hospitalized for COVID-19 in children using the nationwide French National Health Data System (SNDS).

## 2. Materials and Methods

### 2.1. Data Sources

We conducted a French cohort study that compared children vaccinated with the MMR vaccine and those who were not using the SNDS. The SNDS records all in- and out-patient claims, as well as hospital discharges, for 99.5% of the 67 million inhabitants of France. Each individual is given a unique anonymous identifier (encrypted twice in the SNDS) that is valid from birth (or immigration) to death. This identifier is linked to information from the DCIR (Datamart de Consommation Inter-Regimes, the national health insurance re-imbursement database) and the PMSI (Programme de Médicalisation des Systèmes d’Information, the national hospital discharge database).

The DCIR database includes personal data (age, gender, department of residence, vital status) and individual information on out-patient medical care and re-imbursed health products (drugs and vaccines) [17,18,19,20]. Drugs were coded according to the anatomical therapeutic chemical (ATC) classification. The health expenditures of patients with severe and costly long-term diseases (LTDs), such as cancer or diabetes, are fully re-imbursed. Their diagnosis is recorded according to the tenth revision of the International Statistical Classification of Diseases and Related Health Problems (ICD-10).

The PMSI database indicates all public and private hospital admissions in France and hospital/discharge diagnoses coded according to the ICD-10. Medical and surgical procedures are coded according to the French Classification of Medical Procedures. These databases were fully described elsewhere and are regularly used for drug monitoring [2,20,21,22,23,24,25,26].

### 2.2. Study Population

All children in metropolitan France aged 12 to 144 months (1–12 years) on 1 March 2020, who received at least one healthcare re-imbursement in the previous two years, regardless of their health insurance affiliation, were included in this study (excluding twins because of the impossibility of differentiating the twin rank in the PMSI data). The lower age threshold of 12 months corresponds to the first dose of the MMR vaccine according to the French vaccination program. The upper limit of 144 months corresponds to the year (2009), in which the children’s data were consolidated. The index date was the date of entry into the cohort, i.e., 1 March 2020. We excluded children whose vaccination status was uncertain (i.e., children without any health benefits in the first year of life and those with a consultation at a maternal and child protection center) from the non-exposed population to avoid a mis-classification of vaccine exposure. We excluded children who received mono-valent measles or rubella vaccines and those vaccinated with MMR during follow-up (N = 127,948). The cohort was followed from 1 March 2020 to 31 August 2021.

### 2.3. Exposure to the MMR Vaccine

Exposure was defined using MMR vaccination re-imbursements any time from birth to 1 March 2020 (ATC code J07BD52, Appendix A). The date of the first re-imbursement of the MMR vaccine was considered as the date of exposure, identified through its code in the DCIR. Non-exposed children came from the same source population as the exposed children, but for whom no claim nor other information regarding the MMR vaccine was found.

### 2.4. Outcomes

#### 2.4.1. Main Outcome: Hospitalization for COVID-19

Our study was based on data reported from 1 March 2020 to 31 August 2021. The main outcome of the study was hospitalization for COVID-19. It consisted of hospital stays coded using the following ICD-10 codes as the main diagnosis: U07.10 “COVID-19, respiratory form, virus identified”; U07.11 “COVID-19, respiratory form, unidentified virus”; U07.14 “COVID-19, other clinical forms, identified virus”; U07.15 “COVID-19, other clinical forms, virus not identified”; and U08.9 “Personal history of COVID-19, unspecified”.

#### 2.4.2. Secondary Outcome: Hospitalization for COVID-19 or PIMS

Over the 2020/2021 period, cases of pediatric multi-systemic inflammatory syndrome (PIMS) associated with or post-SARS-CoV-2 infection were diagnosed in French hospitals [27,28], leading us to consider them as an additional outcome. PIMS cases were defined by the ICD-10 codes used as the associated diagnosis for multi-system inflammatory syndrome (U10.9) in 2020/2021 and as the main diagnosis for Kawasaki syndrome (M30.3), myocarditis (I40-I41-I51.4), and cardiogenic shock (R57.0) in 2020.

### 2.5. Co-Variates

Co-variates used to evaluate the association between exposure to the MMR vaccine and the risk of COVID-19 hospitalization were potential confounders, and their definition was based on the known literature [29,30]. The following factors were considered: socio-demographic and economic data, such as age, sex, region of residence determined by the telephone codes in France, de-limiting five geographical areas, the universal health coverage (CMU) benefits, and social deprivation index [31]; chronic conditions (Appendix A) recognized to be strongly related to COVID-19 based on the pediatric chronic conditions classification developed by Feudtner et al. [32,33]; the overall status of health indicated through the consumption of healthcare, i.e., history of hospitalizations (excluding birth-associated stays and hospitalizations of less than one day) and consultations with a general practitioner (GP) or pediatrician (excluding screening, mandatory check-ups, and psychiatric visits) in the 24 months prior to 1 March 2020; and exposure to at least one other vaccine (*Haemophilus influenzae* B, meningococcus, pertussis, pneumococcus, tetanus, other bacterial vaccines, influenza, hepatitis B, hepatitis A, diphtheria, poliomyelitis, or papillomavirus).

### 2.6. Statistical Analyses

Descriptive statistics comparing the characteristics of children exposed and not exposed to the MMR vaccine were obtained. Chi [2] tests were used to analyze categorical variables and Student’s *t*-test for quantitative variables with a normal distribution (Wilcoxon’s test for those deviating from normality). The association between exposure to the MMR vaccine and COVID-19 hospitalization outcomes was calculated using a non-conditional logistic regression model. Crude and adjusted odds ratios (ORs) and corresponding 95% confidence intervals were estimated. Confounders for the multi-variate analysis consisted of all variables significantly associated with MMR vaccination and hospitalization for COVID-19 (age, sex, CMU, region, social deprivation index, previous consultations and hospitalization, and co-morbidities). We used propensity score-based methods to account for non-randomized MMR vaccine exposure and to reduce the effects of confounders. For the main analysis, we used the inverse probability of treatment weighting (IPTW) to consider propensity scores. Individual probabilities of MMR vaccine exposure were estimated using a multi-variate multi-nomial logistic regression model and included significantly associated co-variates in the final multi-variate model [34]. The predicted probabilities from the propensity score model were used to calculate the stabilized inverse probability of treatment weighting. We assessed the balance of individual co-variates before and after the inverse probability of treatment weighting. Standardized differences were calculated as the differences in means or proportions divided by the pooled standard deviation. A negligible difference was defined as an absolute standardized difference of <0.1. Other sensitivity and sub-group analyses were performed to strengthen the robustness of our main analyses.

All statistical tests were two-sided, with a type I error of 5%. Statistical analyses were performed using SAS Enterprise Guide 7.1 (SAS Institute, Cary, NC, USA).

## 3. Results

### Descriptive and Analytic Results

During the study period, we identified 7,184,704 children aged 1 to 12 years, of whom 6,800,542 were exposed to the MMR vaccine and 384,162 were not exposed (Figure 1). Approximately 98% had at least one re-imbursement for another vaccine (96%, 94%, and 81% for diphtheria–tetanus–pertussis–polio–*Haemophilus influenzae* b–Hepatitis B, pneumococcus, and meningococcus, respectively). The median age at the time of the first MMR shot was 12 months (interquartile range 11–14 months); 58% were prescribed by a GP and 36% by a pediatrician.

Baseline characteristics according to MMR exposure are presented in Table 1. The median age for the entire population was six years IIQ [3,4,5,6,7,8], with 51% being female. Sixteen percent of children were covered by their parents’ CMU. Approximately 82% had at least one previous consultation with a GP or pediatrician and 4.6% had at least one hospitalization in the previous two years. The most common co-morbidities were chronic respiratory and psychiatric diseases (8% and 1%, respectively). The overall distribution of socio-economic and demographic characteristics and co-morbidities differed minimally according to MMR vaccine exposure, except for respiratory diseases, which were more frequent among children exposed to the vaccine (8.3% versus 4.7%). In addition, children exposed to the MMR vaccine were more likely to be seen by a GP or pediatrician than those who were not exposed.

Between 1 March 2020 and 31 August 2021, 911 children were hospitalized for COVID-19, out of whom 873 (96%) had been exposed to the MMR vaccine. The distribution curves of COVID-19 hospitalization over the entire study period are presented in Figure 2.

The median age of children hospitalized for COVID-19 was five years IIQ [2,3,4,5,6,7,8], and 45% were female (Appendix A). The children most often hospitalized were the youngest, under the age of 10. Their parents had a higher social deprivation index (20% and 28% with a level of four and five, respectively) and 32% were covered by the CMU versus 16% in the non-hospitalized group. Approximately 80% of the hospitalized children had a history of more than three visits to a GP or pediatrician, versus 70% of those that were not hospitalized, and 23% had at least one hospitalization in the previous 24 months, versus 5% in the non-hospitalized group. The most common co-morbidities were respiratory (20% vs. 8%), neurological or neuromuscular (6% vs. 1%), and hematological or immunological (6% vs. 0.2%).

The adjusted odds ratios (aORs) of the association between hospitalization for COVID-19 and MMR vaccine exposure was estimated to be 1.19 [0.85–1.65] (Table 2). We selected all variables significantly associated with hospitalization for COVID-19 from this multi-variate model to calculate the propensity scores (Appendix A). Using the IPTW method, an aOR = 1.09 [0.81–1.48] was estimated. The sensitivity and sub-group (sex and age) analyses are presented in Table 3. Excluding children exposed after 1 January 2018 (start of MMR vaccination requirement) resulted in an estimated aOR = 1.10 [0.79–1.54], and those with co-morbidities had an aOR = 1.30 [0.88–1.93]. The exploration of the factors associated with the risk of hospitalization for COVID-19 identified that children aged 10 to 12 years had a higher risk than children aged 1 to 4 years (aOR = 1.29 [1.03–1.61]) and those who were CSS beneficiaries (aOR = 1.90 [1.64–2.19]). A higher social deprivation index (fifth vs. first quintiles: aOR = 1.81 [1.45–2.26]; fourth vs. first quintiles: aOR = 1.55 [1.23–1.96]), a proxy of general health, such as a history of hospitalizations (aOR = 3.91 [3.30–4.63]) or more than one visit per month (13 or more vs. 1–3: aOR = 1.39 [1.16–1.65]) and co-morbidities (aOR = 2.73 [2.34–3.18]) were factors that increased the risk of hospitalization for COVID-19.

## 4. Discussion

Our study was the first to involve seven million children, the entire pediatric population of the country, to evaluate the effectiveness of the MMR vaccine on the risk of hospitalization for COVID-19. Although this pediatric population is the most affected by MMR vaccination (and other childhood vaccines), we did not observe any significant associations.

Several studies have previously reported on the protective effect of MMR vaccination on COVID-19 outcomes in children and adults [9,15,35]. Most evaluated was the impact of MMR vaccination on the risk of SARS-CoV-2 infection and not hospitalization. The available COVID-19 vaccines only protect against severity and not infection. Hospital data were, therefore, essential to assess the impact of a vaccine (here, MMR) in children, as they experience fewer severe cases and are less affected by COVID-19 vaccines.

There are a number of possible explanations for the non-association or non-protective effect of MMR vaccination on the risk of hospitalization for COVID-19 found in this study. The absence of social interactions due to school closures resulted in the low exposure of children to the virus, and the existence of the effect of any other vaccine that could have had a synergistic action explains the failure to demonstrate an impact of MMR vaccination alone. During their growth, children are most often subject to various respiratory diseases that stimulate the production of antibodies; these antibodies are likely to protect them against future respiratory disorders. There may also be no cross-immunity from the MMR vaccine. Multiple visits to a GP or pediatrician and at least one hospitalization prior to our study period increased the risk of hospitalization for COVID-19. These characteristics were considered to be a proxy that reflected a patient’s fragile general condition. In addition, being a CMU beneficiary or having a high social deprivation index (i.e., living in a dis-advantaged city) increased the risk of hospitalization. This result was consistent with that of the study of Semenzato et al. on the entire French population. As reflected in our analyses, previous studies also reported an increased risk for children with at least one co-morbidity [2,29,36].

This study had a number of strengths. First, it was the first real-life study to evaluate the impact of a live attenuated vaccine on the risk of COVID-19 hospitalization. The SNDS is a medico-administrative database that allowed us to conduct, to date, one of the largest studies to explore the impact of MMR vaccination during the pandemic. In particular, access to these data allowed us to perform an exhaustive cohort study, limiting selection bias. Second, this study covered the period of the alpha and delta strains, which are the most severe variants of COVID-19. Third, we found the same MMR vaccine coverage estimates between 2015 and 2018 (84.5%, 84.9%, 87.5%, and 88.8%, respectively) as for those reported on in the first annual review of infant vaccination obligation [37], validating the criteria for selecting our population, as the estimates were determined using the same database. In France, most vaccines were prescribed by GPs, as they are the most accessible to families because of their country-wide distribution. Pediatricians are fewer in number and concentrated in large cities. Fourth, the most frequent chronic diseases found in our population were respiratory and psychiatric diseases. These results were consistent with what has previously been reported in the literature; particularly, asthma and allergic pathologies are in the forefront of pediatric co-morbidities [29,38,39,40,41]. We noted a higher frequency of these pathologies in vaccinated than non-vaccinated children. This difference was explained by the fact that vaccinated children also benefitted from regular medical follow-ups (frequent consultations and early diagnoses). Fifth, in this study, the children most often hospitalized for COVID-19 were the youngest, as reported in clinical practice [42]. Indeed, the child’s immune system develops from birth and most often matures at approximately three years of age. The fragility of their immune defenses can explain the increase in hospitalization in this category. A number of studies reported a higher median age [30,43,44]. This difference could be explained by our inclusion criteria, which focused on children aged 1 to 12 years, whereas these studies considered children aged under 18 years. Furthermore, these cohorts were composed of both hospitalized children and those who were not. In addition to being younger, male children were more often hospitalized, a result corroborated by other studies [41,42,45]. However, this difference was not significant. The increase in cases in March–April 2020 corresponded to the first wave in France, and the sub-sequent drop in the curves could be explained by the impact of the strict containment procedures enacted by the government from mid-March to May 2020. The increase in hospitalizations in September corresponded to the resumption of the pandemic in France (second wave from September to November 2020). The above results, therefore, reflected the overall representativeness of the selected population and the validity of our analyses.

Our study also had several limitations. First, our study was not able to assess the association between MMR vaccine exposure and hospitalization attributable to COVID-19 from a primary base population consisting of children with a SARS-CoV-2-positive test. Data on laboratory SARS-CoV-2 test results were not available for children (they were tested under their parents’ identifiers). This selection bias could have led to an under-estimation of the proportion of hospitalizations attributable to COVID-19 among both the exposed and non-exposed children, thus, making it difficult to estimate which way the results would advance. Second, the absence of biological and clinical data in the SNDS did not allow us to consider this information in our analyses. A residual confounding effect from un-measured co-variates may, therefore, remain. However, we believe this would not have changed our results, given the methodological safeguards and validation performed to ensure the robustness of our results.

## 5. Conclusions

This study provided arguments that mitigated the hypothesis concerning the cross-protection of live attenuated virus vaccines against severe COVID-19. However, we could not draw any causal conclusions. It would be impossible to conduct a randomized study with an arm without MMR vaccination.

## Figures and Tables

**Figure 1 vaccines-10-01938-f001:**
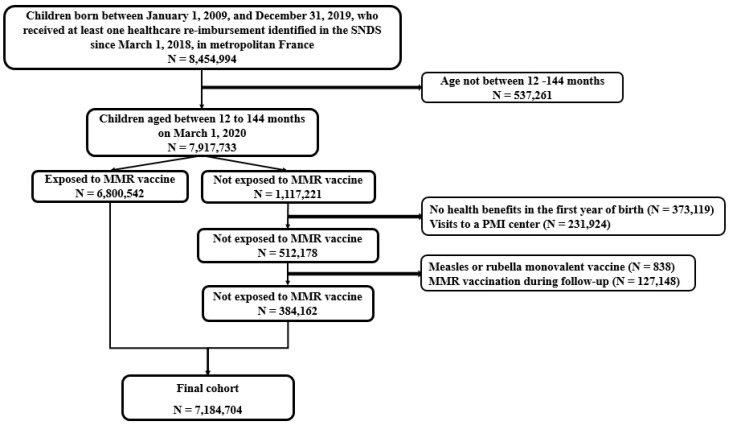
Study flowchart.

**Figure 2 vaccines-10-01938-f002:**
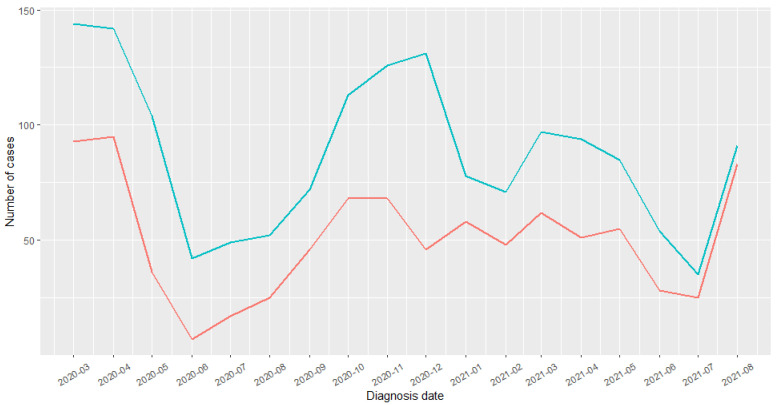
COVID-19 diagnoses distribution. Blue: children were hospitalized for COVID-19; Red: out of whom had been exposed to the MMR vaccine.

**Table 1 vaccines-10-01938-t001:** Baseline characteristics according to MMR vaccine exposure group.

	Exposed to MMR Vaccine N = 6,800,542	Not Exposed to MMR Vaccine N = 384,162	Whole Population N = 7,184,704
**Age (months)—Median (IIQ)**	75 [45–105]	81 [47–111]	75 [45–105]
**Age (years)—Median (IIQ)**	6 [3–8]	6 [3–9]	6 [3–8]
**Age—*n* (%)**			
1–4	2,510,919 (37.0)	125,755 (32.7)	2,636,674 (36.7)
5–9	3,443,691 (50.6)	188,939 (49.2)	3,632,630 (50.6)
10–12	845,932 (12.4)	69,468 (18.1)	915,400 (12.7)
**Female sex—*n* (%)**	3,316,762 (48.8)	189,234 (49.2)	3,505,996 (48.8)
**CMU beneficiaries—*n* (%)**	1,077,298 (15.8)	65,721 (17.1)	1,143,019 (15.9)
**Region—*n* (%)**			
Île-de-France	1,338,383 (19.7)	75,330 (19.6)	1,413,713 (19.7)
Northwest	1,377,208 (20.3)	67,072 (17.5)	1,444,280 (20.1)
Northeast	1,524,092 (22.4)	67,977 (17.7)	1,592,069 (22.2)
Southeast	1,686,970 (24.8)	118,071 (30.7)	1,805,041 (25.1)
Southwest	851,461 (12.5)	53,981 (14.1)	905,442 (12.6)
Unknown	22,428 (0.3)	1731 (0.4)	24,159 (0.3)
**Social deprivation index (quintiles)—*n* (%)**			
1 (the least deprivation)	1,362,638 (20.0)	72,457 (18.9)	1,435,095 (20.0)
2	1,376,843 (20.2)	76,104 (19.8)	1,452,947 (20.2)
3	1,323,056 (19.5)	78,690 (20.4)	1,401,396 (19.5)
4	1,291,105 (19.0)	78,690 (20.5)	1,369,795 (19.1)
5 (the most deprivation)	1,320,012 (19.4)	73,171 (19.0)	1,393,183 (19.4)
Unknown	126,888 (1.9)	5400 (1.4)	132,288 (1.8)
**Exposed to other vaccines—*n* (%)**	6,752,998 (99.3)	313,940 (81.7)	7,066,938 (98.4)
**Number of previous consultations with a GP or pediatrician—*n* (%)**			
No consultation	447,194 (6.6)	112,181 (29.2)	559,375 (7.8)
1–3	1,345,416 (19.8)	105,901 (27.6)	1,451,317 (20.2)
4–7	1,913,406 (28.1)	86,873 (22.6)	2,000,279 (27.8)
8–12	1,512,152 (22.2)	48,526 (12.6)	1,560,678 (21.7)
13 or more	1,582,374 (23.3)	30,681 (8.0)	1,613,055 (22.5)
**At least one hospitalization—*n* (%)**	316,120 (4.6)	14,054 (3.6)	330,174 (4.6)
**Co-morbidities—*n* (%)**			
Neurological/neuromuscular	52,843 (0.8)	2889 (0.7)	55,732 (0.8)
Respiratory	562,916 (8.3)	18,268 (4.7)	581,184 (8.1)
Cardio-vascular	220,806 (0.3)	995 (0.2)	21,801 (0.3)
Psychiatric	87,895 (1.3)	4340 (1.1)	92,235 (1.3)
Metabolic	13,897 (0.2)	800 (0.2)	14,697 (0.2)
Gastro-intestinal	8169 (0.1)	528 (0.1)	8697 (0.1)
Renal/urological	12,985 (0.2)	539 (0.1)	13,524 (0.2)
Hematological/immunological	14,454 (0.2)	819 (0.2)	15,273 (0.2)
Genetic/congenital defect	19,376 (0.3)	976 (0.2)	20,352 (0.3)
Cancer	5007 (0.1)	301 (0.1)	5308 (0.1)

**Table 2 vaccines-10-01938-t002:** Associations between exposure to MMR vaccine and COVID-19 hospitalization outcomes.

	COVID-19	COVID-19 or PIMS
No. of events/no. of individuals	911/7,184,704	1580/7,184,704
Crude analysis—OR (95%CI)	1.29 [0.93–1.79]	1.16 [0.92–1.47]
Multi-variate logistic regression model *—aOR (95%CI)	1.19 [0.85–1.65]	1.05 [0.83–1.34]
Propensity score approach		
Inverse probability of treatment weighting (IPTW)—aOR (95%CI)	1.09 [0.81–1.48]	1.03 [0.82–1.29]

* Adjusted for age, sex, CMU, region of residence, social deprivation index, previous consultations, previous hospitalizations, and co-morbidities.

**Table 3 vaccines-10-01938-t003:** Sensitivity and stratified analysis of the association between exposure to MMR vaccine and the risk of hospitalization for COVID-19 in a multi-variate logistic regression model weighting for the inverse probability of treatment.

	No. of Events/No of Individuals	aOR (95%CI) *
**Sensitivity analysis**		
Exposed to MMR vaccine before 1 January 2018	626/5,659,103	1.10 [0.79–1.54]
Exposed to at least two doses of MMR vaccine	717/6,034,653	1.07 [0.78–1.46]
Exposed to up to two doses of MMR vaccine	623/5,447,486	1.06 [0.77–1.46]
Exclusion of children aged one year	776/6,617,726	0.98 [0.72–1.35]
Exclusion of children with at least one comorbidity	614/6,414,384	1.30 [0.88–1.93]
**Stratified analysis**		
Stratification by sex		
Male	501/3,678,708	0.96 [0.65–1.42]
Female	410/3,505,996	1.31 [0.80–2.14]
Stratification by age		
1 to 4 years	397/2,636,674	1.03 [0.64–1.66]
5 to 9 years	399/3,632,630	1.38 [0.82–2.31]
10 to 12 years	115/915,400	1.11 [0.54–2.29]

* Co-variates used for inverse probability of weighting: age, sex, CMU, region of residence, social deprivation index, previous consultations, previous hospitalizations, and co-morbidities.

## Data Availability

According to data protection and French regulations, the authors cannot publicly release the data from the French National Health Data System (SNDS). However, any person or organization (public or private; for-profit or non-profit) can access anonymized SNDS data to perform a study, research, or an evaluation of public interest, upon authorization from the French Data Protection Office (Available online: https://www.snds.gouv.fr/SNDS/Processus-d-acces-aux-donnees and https://documentation-snds.health-data-hub.fr/introduction/03-acces-snds.html, accessed on 1 January 2021).

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
