# Peer review of "Does Measles, Mumps, and Rubella (MMR) Vaccination Protect against COVID-19 Outcomes: A Nationwide Cohort Study"

_vaccines, 2022, doi:10.3390/vaccines10111938_

Round 1
Reviewer 1 Report
It appears that the research has been planned in a very correct way, involving an impressive number of subjects and obtaining the data from official, well-known sources: SNDS (in and outpatients claims + hospital discharge), DCIR (National Health Insurance Reimbursement) and PMSI (hospital discharge database) considering the limitations of all of them (eg: no lab and clinical tests for the SNDS).
The statistical tests employed are described in detail, including the discussion on confounders, and appear to have been correctly used. I also appreciated the evaluation of the different events which could have (but didn’t) canceled the effective protection offered by MMR against hospitalization due to Sars-Cov-2.
In the Conclusions, the Authors affirm that the results of their work do not confirm the hypothesis that the MMR vaccination, performed with live attenuated viruses, protects from SARS-CoV-2 hospitalization; but, intellectually honest, they admit that a true “causal” conclusion cannot be reached because it would be too difficult - or better, impossible – “to conduct a randomized study with an arm without MMR vaccination”.
Author Response
Comments and Suggestions for Authors
It appears that the research has been planned in a very correct way, involving an impressive number of subjects and obtaining the data from official, well-known sources: SNDS (in and outpatients claims + hospital discharge), DCIR (National Health Insurance Reimbursement) and PMSI (hospital discharge database) considering the limitations of all of them (eg: no lab and clinical tests for the SNDS).
Response 1: Thank you so much for your comment.
The statistical tests employed are described in detail, including the discussion on confounders, and appear to have been correctly used. I also appreciated the evaluation of the different events which could have (but didn’t) canceled the effective protection offered by MMR against hospitalization due to Sars-Cov-2.
Response 2: Thank you so much for your comment.
In the Conclusions, the Authors affirm that the results of their work do not confirm the hypothesis that the MMR vaccination, performed with live attenuated viruses, protects from SARS-CoV-2 hospitalization; but, intellectually honest, they admit that a true “causal” conclusion cannot be reached because it would be too difficult - or better, impossible – “to conduct a randomized study with an arm without MMR vaccination”.
Response 3: Thank you for your comment.
Reviewer 2 Report
Major Points:
- The authors cite a study that evaluates possible cross-protection of MMR vaccination against SARS-CoV-2 infection (Lundberg et al. 2021) as a premise and setup to their study. However, in that report, the protection effects are measured soon after MMR vaccination, since this one in the context of a measles outbreak. The other should take into account how much time has passed time since the first or second doses of MMR vaccination when drawing conclusions for cross-protection.
- Another difference to the Swedish study they cite, is that in Lundberg et al. study the conclusions are drawn against "testing positive" for SARS-CoV-2 infection. The metric here, hospitalizations, does not capture this effect - if any. Even though the authors declared this as a limitation to their study, I wonder if they have access to the COVID diagnosis status of the children's parents soon after MMR vaccination. In other words, how likely were recently vaccinated children to become hospitalized after their parents tested positive for SARS-CoV-2? Are we missing this exposure window?
- MMR vaccination has been previously linked with low severity of COVID-19 (Ashford et al. 2021). Severity, length of treatment and mortality are not factored into this study. Is this information available? Could a possible protective effect be being missed here due to the lack of this information?
- Can the authors separate effects from MMR alone and other pediatric vaccines (diphteria, poliomyelitis)?
Minor Comments:
Line 39 - in the Hubei Province
Line 58 - Confusing sentence; please re-write.
Line 199 - Please define aOR in text not just abstract.
Author Response
Comments and Suggestions for Authors
Major Points:
- The authors cite a study that evaluates possible cross-protection of MMR vaccination against SARS-CoV-2 infection (Lundberg et al. 2021) as a premise and setup to their study. However, in that report, the protection effects are measured soon after MMR vaccination, since this one in the context of a measles outbreak. The other should take into account how much time has passed time since the first or second doses of MMR vaccination when drawing conclusions for cross-protection.
Response 1: Thank you for this question.
Lundberg et al. 2021 assessed the effect of recent MMR vaccination (2018) on the risk of SARS-CoV-2 infection in health care workers.
Indeed, we performed sensitivity analyses based on the same approach. We restricted the analysis to children who had been vaccinated in the 12 and 24 months before the index date., i.e., years 2019 and 2018, respectively, to assess whether a recent vaccination was associated with the risk of hospitalization for COVID-19. The results of this investigation showed the absence of cross-protection of the MMR vaccine against hospitalisation for SARS-CoV-2 infection (adjusted odds ratio of 1.21 [0.91-1.62] and 1.13 [0.85-1.51] for first or second dose 12 months and 24 months before the index date, respectively).
We added these results to Supplementary Table S2.
- Another difference to the Swedish study they cite, is that in Lundberg et al. study the conclusions are drawn against "testing positive" for SARS-CoV-2 infection. The metric here, hospitalizations, does not capture this effect - if any. Even though the authors declared this as a limitation to their study, I wonder if they have access to the COVID diagnosis status of the children's parents soon after MMR vaccination. In other words, how likely were recently vaccinated children to become hospitalized after their parents tested positive for SARS-CoV-2? Are we missing this exposure window?
Response 2: Thank you for this question. This would have been a great analysis to perform. However, we do not have the data to perform it:
- Indeed, In the SNDS database, children and parents are each distinguished by a single anonymous identifier. Thus, a parent and his or her children appear as separate subjects and the filiation cannot be tracked due to anonymisation.
- No COVID-19 test data (screening and diagnosis lab tests) are available in the SNDS database for adults or children.
- MMR vaccination has been previously linked with low severity of COVID-19 (Ashford et al. 2021). Severity, length of treatment and mortality are not factored into this study. Is this information available? Could a possible protective effect be being missed here due to the lack of this information?
Response 3: Thank you for this question.
The main outcome in our study is the risk of hospitalisation for COVID-19, considered to be a marker of the severity of SARS-CoV-2 infection. We clarified this point in the conclusion by adding "severe" to qualify the outcome (Line 309).
We were not able to assess the impact of MMR vaccination on the risk of ICU admissions and mortality from COVID-19 due to the small size of the samples (73 children were admitted to the ICU and one died (Supplementary Table S1)).
- Can the authors separate effects from MMR alone and other pediatric vaccines (diphteria, poliomyelitis)?
Response 4: Thank you for this question.
We cannot individually separate exposure to other vaccines from exposure to the MMR vaccine. As vaccination against diphtheria and poliomyelitis is mandatory in France, all children are exposed to it from the 2nd month of life. Then, at 12 and 18 months of age, they receive doses of the MMR vaccine. Therefore, children exposed to MMR have already been exposed to these other vaccines and in the context of our study we cannot separate the effect of the MMR vaccine alone from that of the other paediatric vaccines.
Minor Comments:
Line 39 - in the Hubei Province
We thank you for this suggestion. We took it into account in the manuscript line 39.
Line 58 - Confusing sentence; please re-write.
Previous sentence: Scientific evidence that live attenuated vaccines could offer an alternative strategy or prevent this infection could have an impact on public health policy as the pandemic continues, with multiple mutations.
Sentence re-write: Providing scientific evidence that live attenuated vaccines have the potential to prevent infection and/or the severity of Covid-19 could have an impact on public health policy as the pandemic continues.
Line 199 - Please define aOR in text not just abstract.
Thank you for these suggestions. We added “adjusted odds ratios (aOR)” in line 201 of the manuscript.
Round 2
Reviewer 2 Report
All my concerns were addressed.